# Maternal death surveillance and response system evaluation in Makonde District, Zimbabwe, 2021

**Tsitsi Brenda Makanyanga[1,2], Bernard Madzima[2], More Mungati[3], Addmore Chadambuka[1] \*, Notion Tafara Gombe[4], Tsitsi Patience Juru[1], Chukwuma David Umeokonkwo[5], Mufuta Tshimanga[1]**

**1** Department of Primary Health Care Sciences, Family Medicine, Global and Public Health Unit, University of Zimbabwe, Harare, Zimbabwe, **2** National AIDS Council of Zimbabwe, Harare, Zimbabwe, **3** Elizabeth Glaser Pediatric AIDS Foundation, Harare, Zimbabwe, **4** African Field Epidemiology Network, Harare, Zimbabwe, **5** African Field Epidemiology Network, Monrovia, Liberia

\* achadambuka1@yahoo.co.uk

**Data Availability Statement:** The data underlying the results presented in the study are available from figshare at 10.6084/m9.figshare.23559579.

## Abstract

### Background

Maternal mortality is of global concern, almost 800 women die every day due to maternal complications. The maternal death surveillance and response (MDSR) system is one strategy designed to reduce maternal mortality. In 2021 Makonde District reported a maternal mortality ratio of 275 per 100 000 and only sixty-two percent of deaths recorded were audited. We evaluated the MDSR system in Makonde to assess its performance.

### Methods

A descriptive cross-sectional study was conducted using the CDC guidelines for evaluating public health surveillance systems. An Interviewer-administered questionnaire was used to collect data from 79 health workers involved in MDSR and healthcare facilities. All maternal death notification forms, weekly disease surveillance forms, and facility monthly summary forms were reviewed. We assessed health workers' knowledge, usefulness and system attributes.

### Results

We interviewed 79 health workers out of 211 workers involved in MDSR and 71 (89.9%) were nurses. The median years in service was 8 (IQR: 4–12). Overall health worker knowledge (77.2%) was good. Ninety-three percent of the deaths audited were of avoidable causes. Twelve out of the thirty-eight (31.6%) facilities were using electronic health records system. Feedback and documented shared information were evident at four facilities (21%) including the referral hospital. Nineteen (67.9%) out of 28 maternal death notification forms were completed within seven days and none were submitted to the PMD on time.

**Funding:** The author(s) received no specific funding for this work.

**Competing interests:** The authors have declared that no competing interests exist.

## Conclusion

The MDSR system was acceptable and simple but not timely, stable and complete. Under-utilization of the electronic health system, work load, poor documentation and data management impeded performance of the system. We recommended appointment of an MDSR focal person, sharing audit minutes and improved data management.

## Introduction

Maternal mortality is of global concern, almost 800 women die every day due to preventable maternal complications [1]. Maternal mortality remains a challenge in Sub-Saharan Africa and Zimbabwe is not spared. Maternal mortality ratio of Sub Saharan Africa was 542 compared to a global ratio of 216 [2]. In Zimbabwe, according to the ZIMSTATS Multi-cluster Survey done in 2019 the maternal mortality ratio was recorded as 462 per 100 000 live births [3]. Globally the maternal mortality ratio is declining however the average rate of decline is not on track to achieve United Nations Sustainable Development Goal target 3.1 of reducing maternal mortality ratio to 70 maternal deaths per 100000 live births by 2030 and that no country should have an maternal mortality ratio greater than 140 [4]. These set targets have formalized the efforts of the fight towards ending maternal mortality [5]. The United States Centre for Disease Control Division of Reproductive Health collaborated with the World Health Organization (WHO) to develop Maternal Death Surveillance and Response guidelines as one strategy to achieve the goal to reduce maternal mortality [6]. The guidelines were launched in 2013 and Zimbabwe adopted the MDSR the same year [7].

A Maternal Death Surveillance and Response (MDSR) system is a continuous audit cycle of maternal death identification, collection of data, investigation, notification and review followed by interpretation of findings, response, recommendations and actions to prevent future deaths and evaluation of impact of public health actions implemented [6]. The MDSR system stipulates that a maternal death be treated as a notifiable event and incorporated into a country's reporting system for notifiable diseases [8]. The specific objectives of the MDSR system are [6]:

1. Collect accurate data on all maternal deaths that are the actual number, causes of deaths and contributing factors.

2. Analyze and interpret the data to determine trends and patterns in maternal mortality, determine contributing factors and determine whether deaths were avoidable or not.

3. Make evidence-based recommendations using data collected.

4. Dissemination of findings and recommendations to the civil society, health personnel, policy makers and decision makers.

5. Monitor and evaluate the implementation of recommendations to reduce maternal mortality.

6. Inform on effectiveness and impact of interventions implemented to reduce maternal mortality.

7. Allocate resources more effectively and efficiently.

8. Enhance accountability of maternal health.

9. Improve maternal mortality statistics.

10. Guide and prioritize research related to maternal mortality.

11. Demonstrate accountability to clients.

Makonde District recorded a maternal mortality ratio of 275 in 2021. Review of the MDSR system in Makonde District showed delays in notification of maternal deaths to the provincial medical directorate (PMD) and recording discrepancies. Only sixty-two percent of maternal deaths were notified on time and audited. Among the seven districts in Mashonaland West Province, Makonde recorded the highest number of maternal deaths with the least number of audits performed in 2021. The remaining six districts in the province achieved 100% notification and auditing of maternal deaths save for one district which achieved 80% with the second highest number of maternal deaths.

As nations progress towards ending maternal mortality, all strategies and each component of the strategies need to be strengthened in order to achieve the global set targets. The MDSR system if implemented in full capacity will significantly contribute towards ending maternal mortality hence the quality of practice need not be compromised. There are no documented studies that show that the MDSR system of Makonde District has been evaluated. We evaluated the MDSR system in Makonde to assess if it was meeting its intended purpose.

## Materials and methods

### Study setting

The study was done in selected health facilities in Makonde District. Makonde District is one of the seven districts in Mashonaland West Province of Zimbabwe. The projected population of Makonde District for 2021 was 196 771 in total, 102 874 females (52%) and 34 230 women of child bearing age [9]. The population of pregnant mothers was 8363 with eighty-five percent attending at least one antenatal care visit, twenty-three percent attending the antenatal care visit before 16 weeks of pregnancy, thirty-four percent attending the fourth antenatal care visit, forty-seven percent attending the post-natal care visit at six weeks and none had attended eight or more ante-natal care visits as stipulated by WHO. Institutional maternal mortality ratio for Makonde District was 275 per 100 000 live births (Mashonaland West report, 2021).

Makonde District has forty-five registered health facilities which include twenty-eight rural clinics, fourteen urban clinics comprising of five institutional clinics and three private clinics, two mission rural hospitals and one provincial hospital (Chinhoyi) which serves as the referral hospital for Mashonaland West Province. Thirty-eight out of the forty-five health facilities offer maternity services. There is one private maternity clinic in the district. Of these institutions offering maternity services, 52.6% of the total deliveries in the district were from the provincial referral hospital, 28.8% from the 28 rural clinics, 14.8% from one mission hospital, 2.4% from the other mission hospital, 1.2% from the private maternity hospital and 0.2% from the urban council clinics. Chinhoyi Provincial Hospital offers a full package of comprehensive emergency obstetric and newborn care. Every maternal death is reported and investigated through the MDSR system. The number of deliveries and maternal deaths at the facilities in Makonde is shown in Table 1.

**Maternal death surveillance and response system in Zimbabwe.** Any maternal death that occurs in a health facility triggers review. Deaths occurring in health facilities should be notified to the appropriate authorities within twenty-four hours and deaths in communities within forty-eight hours. Active surveillance is part of the MDSR system whereby health facilities and communities should include zero reporting in their routine reports which means

**Table 1. Distribution of deliveries and maternal deaths among participating health facilities in Makonde District, Zimbabwe, 2021.**

| Health Facility | Deliveries | Maternal Deaths |
|---|---|---|
| Chinhoyi Provincial Hospital | 6920 | 22 |
| Makonde Christian Hospital | 1915 | 0 |
| Mtala Council Clinic | 537 | 1 |
| Kanyaga Rural Health Centre | 408 | 0 |
| St Ruperts Mission Hospital | 314 | 0 |
| Umboe Rural Health Centre | 314 | 0 |
| Manyamba Rural Health Centre | 195 | 1 |
| Sadoma Council Clinic | 193 | 0 |
| Shackleton Council Clinic | 193 | 0 |
| Kenzamba Rural Health Centre | 173 | 0 |
| Kamhonde Rural Health Centre | 171 | 0 |
| Green Valley Rural Health Centre | 163 | 0 |
| Chitambo Private Hospital | 155 | 0 |
| River Ranch Rural Health Centre | 139 | 0 |
| Municipal Urban Clinics | 26 | 0 |
| Total | 11816 | 24 |

absence of the event is recorded for the review period [1]. A maternal death notification form is completed in triplicate within seven days, one copy remains at the facility; two copies are forwarded to the PMD within fourteen days and one copy forwarded to the head office within 30 days. Maternal death audits are done at facility level and district level where deaths are reviewed with confidentiality. Case specific recommendations are made during the audits and at each level feedback and dissemination of information must be done. The audit meetings are done monthly at district hospitals, quarterly at provincial level and annually at national level. The maternal mortality trends are best determined in quarterly meetings at provincial level and annually at national level. The district audit is the facility-based audit, defined as the qualitative in-depth investigation of the causes of and circumstances surrounding a maternal death to identify any avoidable factors that could be changed to improve maternal care. It is the level where changes can be made with the greatest impact. At national level, reviews are done and recommendations are feedback to the PMD and the PMD feeds back to the district. The MDSR is continuously monitored and impact continuously evaluated. The flow of information in a Maternal Death Surveillance and Response system in Zimbabwe is shown in Fig 1.

**Operation of the MDSR in Makonde District, 2021.** When a maternal death occurs at the clinic or hospital, the nurse on duty at the maternity ward notifies the head of institution within 24 hours. The nurses fill in yellow cards (deceased patient cards) as soon as a death occurs. One card is kept at the ward and one card is forwarded to the health information office the following morning. The health information officers use these cards to enter information on the line list. A preliminary investigation is done by nurses and doctors who were on duty at the time of death and the Maternal Death Notification Form (MDNF) is filled in, within twenty-four hours. The MDNF should be filled in completely by seven days and the form forwarded to the health information office for capturing on the District Health Information System (DHIS) 2 platform. The health information officer updates the weekly disease surveillance and the Ministry of Health tally sheet 5 (monthly summary) where in the event of no deaths they report 0 on block 13 on the delivery section. Maternal audits are conducted at district level and the Provincial Medical Directorate is represented.

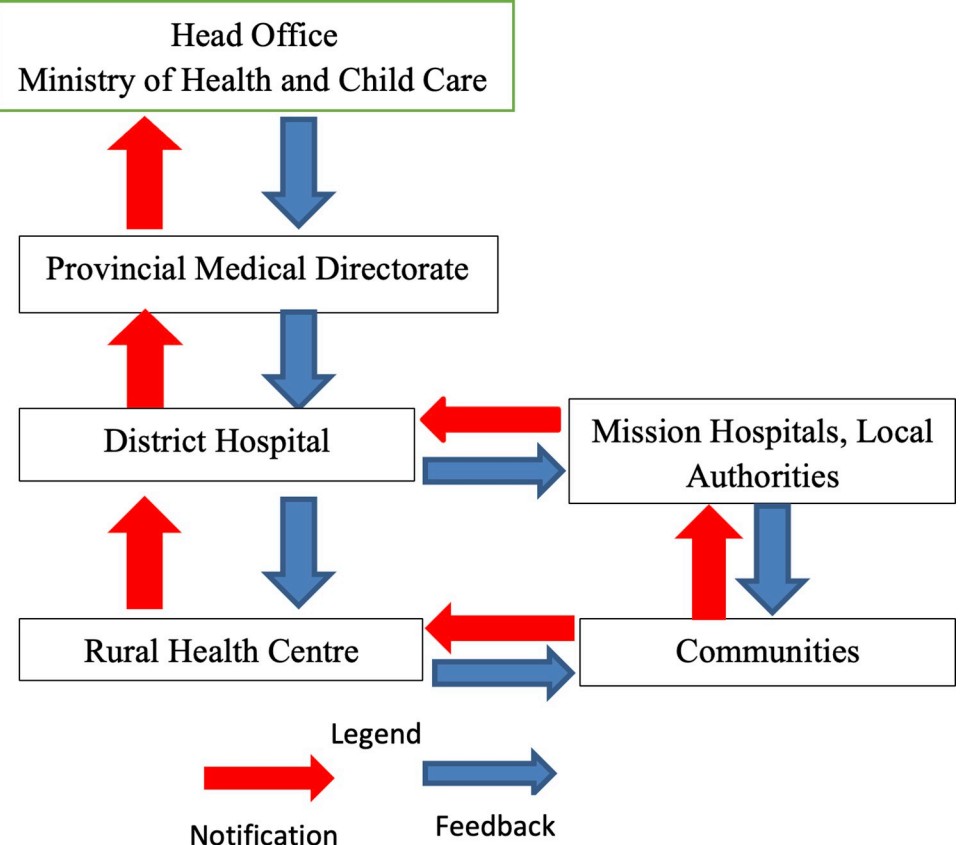

**Fig 1. Flow of information in a maternal death surveillance and response system in Zimbabwe.** (Figure adopted from the Guidelines for Maternal and Perinatal Death Audits in Zimbabwe).

## Study design

We carried out a descriptive cross-sectional study using the Updated CDC Guidelines for Surveillance System Evaluation and specifically the Maternal Death Surveillance and Response Technical Guidance. The surveillance system in Zimbabwe is premised on satisfying attributes that are stipulated in the updated CDC Guidelines. This evaluation assessed the extent to which the surveillance attributes as stipulated in the guidelines were satisfied.

## Study population

The study population was nurses and doctors that operationalize the system. Records reviewed included the monthly summary register which captures all health events, weekly disease surveillance forms, maternal death notification forms and maternal audit reports.

**Sample size.** A minimum sample size of sixty-one health workers was calculated using the Dobson formula, based on a study by Tapesana *et al* [10], where twenty percent of the reported deaths were reviewed. All mortality records available for 2020 and 2021 were reviewed, that is, 28 MDNFs, 28 maternal audit reports, 124 facility monthly summary registers and 104 weekly disease surveillance forms.

**Sampling.** Healthcare workers available at the selected facilities during time of data collection were enrolled in the study to reach the minimum sample size. Nineteen out of the thirty-eight health facilities offering maternity services participated in the study, that is, the only

referral hospital, one of the two mission hospitals, the four urban public clinics, the private maternity clinic and twelve out of the twenty-three rural council clinics were randomly selected to participate in the study. Proportionate sampling based on the establishment in the respective facilities was used to come up with the number of nurses and doctors to be interviewed at the respective facilities. We interviewed five doctors and forty nurses from the provincial hospital, two doctors and six nurses from the rural hospital, one nurse and one doctor from the private maternity clinic, ten nurses from the urban clinics and fourteen nurses from the rural clinics.

## Data collection

Interviewer administered questionnaire was used to collect information of health worker knowledge, usefulness of the surveillance and assess the system's attributes. The questionnaire was created with guidance of monitoring indicators and targets of the MDSR technical guideline [7].

**Measurement of surveillance system attributes, knowledge and usefulness of MDSR among healthcare workers, Makonde District, 2021.** *Timeliness*. We measured timeliness through reviewing records to check whether maternal death notification forms were completed within seven days at the health facility, forwarded to the Provincial Medical Directorate within fourteen days and audits conducted in less than one month.

*Stability*. We assessed ability of the system to provide and manage data without failure through questionnaires and review of records to assess adequacy of manpower, training needs and use of electronic means to manage data.

*Data quality*. Maternal death notification forms were reviewed to check for completeness of forms. The monthly summary and weekly disease surveillance forms were checked for completeness. In the event that there was no death, zero reporting was expected.

*Acceptability*. We assessed willingness of health workers to participate through asking questions and review of records checking for completeness and timeliness.

*Simplicity*. We assessed the attribute by asking respondents about their perception on ease of operating of the system and objective assessment of methods of transmitting case information, training gaps and methods of disseminating surveillance data.

*Knowledge*. A 1–5 Likert scale was used to measure level of knowledge among health care workers with rating categorized as good (4–5), fair (3) and poor (1–2) score. Questionnaires were administered to respondents and knowledge assessed through stating correct definition of MDSR, correct timelines of notifications, objectives of MDSR and name of forms used in the surveillance.

*Usefulness*. We assessed achievements of the system such as public health actions implemented and outcomes. Respondents were asked about their perception on usefulness of the system.

## Data capture and analysis

Data was captured using Epi Info 7.2.4 statistical package. This software was used to conduct univariate analysis to generate frequencies, means and proportions.

## Ethical considerations

The study was reviewed and approved by the Mashonaland West Provincial Ethics Committee (JFREC-NAC/3/22). Written voluntary informed consent was obtained from all participants. The objectives of the study were clearly highlighted to the participants. Privacy and confidentiality were maintained during data collection.

**Permission to proceed.** Permission was sought and obtained from the Ministry of Health and Child Care and Health Studies Office.

# Results

## Demography of healthcare workers

The demographic characteristics of respondents are shown in Table 2. We interviewed seventy-nine healthcare workers. The majority 71 (90%) were nurses. Twenty-nine (41%) of the nurses were midwives. The median years in service was 8 (Interquartile Range (IQR); 4–12).

## Knowledge

Table 3 shows results on knowledge. Overall, 61 (77.2%) of the healthcare workers displayed good knowledge of the MDSR system. Seventy-two (91.1%) respondents knew the timelines of reporting maternal deaths by phone. Sixty-nine (87.3%) of the healthcare workers knew the important tool (MDNF) updated following a maternal death. Seventy percent of the healthcare workers knew the timelines for completing the maternal death notification forms. Ninety percent of the healthcare workers knew that every maternal death must be investigated. Ninety-two percent of the healthcare workers were aware that audits were to be done following a maternal death to come up with recommendations to avoid future deaths of similar avoidable causes.

## Usefulness

Table 4 summarizes the findings on usefulness. Ninety-seven percent of the healthcare workers perceived the system useful. Forty-two (93.3%) deaths recorded on the line list were due to avoidable causes. Twenty-eight (62.2%) deaths were audited with MDNFs completed. Four of the nineteen (21%) smaller health facilities had ever received minutes from the audits done in the district.

**Table 2. Demographic characteristics of healthcare workers in Makonde District, Zimbabwe, 2021.**

| Variable | Frequency n = 79 (%) |
| --- | --- |
| Sex | |
| Male | 56 (70.9) |
| Female | 23 (29.1) |
| Designation | |
| Midwife | 29 (36.7) |
| Registered general nurse | 31 (39.2) |
| Primary care nurse | 11 (14) |
| Medical doctor | 8 (10.1) |
| Median years in service (Interquartile range (IQR)) | 8 (IQR 4–12) |
| Years in service | |
| 1–5 | 22 (27.8) |
| 6–10 | 24 (30.4) |
| 11–15 | 22 (27.8) |
| 16–20 | 7 (8.9) |
| 21–25 | 0 (0) |
| 26–30 | 2 (2.5) |
| 31–35 | 2 (2.5) |

**Table 3. Knowledge of healthcare workers in Makonde District, Zimbabwe, 2021.**

| Variable | Frequency n = 79 (%) |
|---|---|
| Knew the timelines of reporting death by phone | 72 (91.1) |
| Knew the timelines for completing the MDNF | 55 (69.6) |
| Knew that every maternal death must be investigated | 71 (89.9) |
| Knew that audits were done to come up with recommendations to improve future deaths | 73 (92.4) |
| Knew the important tool updated following a maternal death | 69 (87.3) |
| Overall knowledge rating on Likert scale | |
| Poor | 5 (6.3) |
| Fair | 13 (16.4) |
| Good | 61 (77.2) |

## System attributes

**Timeliness.** Table 5 summarizes the findings on timeliness. Nineteen out of 28 MDNFs were completed within seven days and none of the forms were forwarded to the Provincial Medical Directorate within fourteen days. Twelve out of 28 audits were conducted in less than one month.

**Data quality.** Twenty-eight out of 45 (62.2%) expected maternal death notification forms were available. Date of death was captured on 23 of the 28 MDNFs. Section H4 indicating whether death was avoidable or not was completed on all forms reviewed. Section H6 (name of health worker completing the form) was completed on 22 out of 28 MDNFs. Monthly facility registers were completed including zero reporting.

**Stability.** Assessment of stability is shown in Table 6. Forty-four out of 79 (55.7%) healthcare workers reported work overload to deliver the MDSR. Forty-five out of 79 (57%) healthcare workers were trained on MDSR. Twenty-eight out of 45 MDNF (62.2%) were available. Facilities utilizing the electronic health system were 12 out of the 38 facilities (31.6%) in the district. Facility to facility communication breakdown was indicated on 15 out of the 28 MDNFs filled.

**Acceptability.** Seventy (88.6%) respondents were willing to continue to participate in the MDSR system. Almost all (98.7%) except for one respondent did not fear blame and were willing and confident to fill in the MDNFs and attend audits after a maternal death. Majority (96.2%) of the healthcare workers felt it was their duty to notify maternal deaths within the

**Table 4. Usefulness of the MDSR system in Makonde District, Zimbabwe, 2021.**

| Variable | Frequency n (%) |
|---|---|
| Shared information (n = 19) | 4 (21) |
| Deaths due to avoidable causes (n = 45) | 42 (93.3) |
| Perceived usefulness by health care workers | 77 (97) |
| Deaths audited | 28 (62.2) |

**Table 5. Timeliness of the MDSR system in Makonde District, Zimbabwe, 2021.**

| Variable | Frequency n (%) |
|---|---|
| Maternal Death Notification Form (MDNF) completed within seven days (n = 28) | 19 (67.9) |
| MDNF forwarded to Provincial Medical Directorate in fourteen days (n = 13) | 0 (0) |
| Audits conducted in less than one month (n = 28) | 12 (42.9) |

**Table 6. Stability of the MDSR system in Makonde District, Zimbabwe, 2021.**

| Variable | Frequency n (%) |
|---|---|
| Work overload (n = 79) | 44 (55.7) |
| Trained on MDSR (n = 79) | 45 (57) |
| Facilities using the electronic health system (n = 38) | 12 (31.6) |
| MDNF available (n = 45) | 28 (62.2) |
| Audits conducted (n = 45) | 40 (88.9) |
| Facility to facility communication breakdown (n = 28) | 15 (53.6) |

stipulated time. Majority of the respondents were willing to participate and discuss issues concerning maternal health. Six (7.59%) healthcare workers indicated that the surveillance system wasted time as some recommendations were never implemented yet known that they would make an impact in improving maternal health delivery. Ninety-seven percent reported the system not to be time consuming. Majority of healthcare workers, 70 (88.61%) reported the system as not overburdening them. Majority of the healthcare workers (84.8%) reported to be confident to fill in the MDNF when need arises.

**Simplicity.** Forty-one (93.4%) of the respondents had ever filled the maternal death notification form. Thirty-five (85.4%) reported to fill in the form within 20 minutes. Seventy-one (90%) of the respondents indicated that the notification process was easy.

## Discussion

Our study revealed that the maternal death surveillance and response system in Makonde District is functional however there are shortcomings in operation of the system. The system was useful, acceptable to some extent and simple. However, it was not timely, complete and stable.

The healthcare workers in Makonde District displayed good knowledge to meet the objectives of the surveillance system. Majority of the healthcare workers in all facilities including rural health centres were aware of the surveillance system and its objectives, despite that most deaths were recorded from the district referral hospital. These findings were contrary to findings by Tapesana et al, Mutsigiri et al and Maphosa et al [10–12] where healthcare workers from rural facilities displayed poor knowledge of the MDSR system. The key definitions in the surveillance system that include definition of a maternal death and woman of reproductive age were clearly stated by majority of the healthcare workers interviewed. This implies that the healthcare workers can identify maternal deaths and are aware of the target group for surveillance although this does not necessarily translate to good practice.

Healthcare workers in Makonde District presumed the MDSR system to be useful as they utilised the information generated from audits to improve maternal health service delivery and monitor maternal mortality trends. There was evidence of audit minutes and charts addressing issues raised from previous audits. However, to some extent usefulness of the surveillance system was affected by the district performance which was below the expected. Not all audits were done following deaths. Of the audits conducted, some were not done on time to recall events, which resulted in missed opportunities to learn from previous experience which would help improve future outcomes in maternal healthcare. Failure to conduct some audits was possibly attributable to reluctance to fill in maternal death notification forms and failure to fill the forms on time by the health care workers on duty when the maternal death occurs. These findings were also reported by Mwaniki et al [13] in Kenya whereby only fifty-five percent of maternal deaths were audited. Recommendations generated after audits were not circulated to all facilities and stakeholders in the district, they were mostly restricted to facilities linked with

the death. The implementation part of the MDSR system in Makonde was also affected by failure to address some of the recommendations from audits due to lack of resources. Similar findings were also reported by Millimouno et al [14]. The implementation of action plans is the most important and is the response part of the system, hence action plans must be clear and there must be a systematic way of tracking them [15].

Timeliness of reporting in this study was very poor. Poor performance in observing timelines of completing maternal death notification forms was attributed to over workload, reluctance, lack of motivation and missing case information. These findings were also reported by Mutsigiri et al and Maphosa et al [11, 12]. This could also be attributed to failure to realise the importance of completing the maternal death notification forms by some healthcare workers as reported by Said et al [15]. There is need to notify and investigate every death timely enough whilst the memory is still fresh to recall events. This would also improve usefulness of the system.

The acceptability of the MDSR system among healthcare workers was reportedly high although the actual performance of the MDSR system indicated otherwise for example non-review of some maternal deaths and failure to complete maternal death notification forms. Lack of motivation and reluctance by healthcare workers are possible reasons that affected acceptability of the MDSR in Makonde, as similarly reported by Maphosa et al and Bukassa et al [12, 16].

The MDSR system in Makonde District was not complete as evidenced by missing records and missing information on the completed maternal death notification forms. Some healthcare workers still fear blame and are not comfortable to fill in the MDNF or they fill in the forms and not disclose identity despite audits being not fault-finding missions. A study by Tapesana et al also reported similar findings [10]. The healthcare workers highlighted that they failed to access case information in some instances especially for referred cases. Despite introduction of the electronic health system majority of the facilities continue to rely on paper-based communication which may compromise relay of information. Completeness of data could also be attributable to lack of training. The study revealed training gaps among healthcare workers including the health information officers who manage data.

Stability of the system in Makonde was mainly affected by poor data management as was also highlighted by Said et al [15]. Stability was also affected by shortage of staff in the health facilities and retention of midwives as a substantial number of health care workers transferred or left the health care system. It was not uncommon to find general nurses working in the maternity wards as there were not enough midwives to cover the wards and most of the rural clinics did not have any midwives.

## Limitations of the study

Fewer midwives participated in the study, given they are the professional grouping mandated to handle pregnant women limits the validity of information obtained from the health workers, however all the nurses interviewed were working in the maternal and child health department. We failed to reach the desired sample size for village health workers hence the representativeness of the surveillance system could not be ascertained.

## Conclusion

The MDSR system was acceptable and simple but not timely, stable and complete. The system was useful to some extent useful and healthcare worker knowledge was good. Inadequate human resources, poor documentation and poor information flow from one level to another hampered operation of the MDSR system. The MDSR if implemented in full capacity will

significantly contribute towards ending maternal mortality hence the quality of practice need not be compromised.

## Recommendations

We recommended on job training and continued education to health information officers who manage the data, formal documentation of recommendations and sharing of audits with all facilities and stakeholders in the district, functionalising the electronic health system in all facilities and appointing an MDSR focal person to coordinate the system in the district. Training of more midwives is recommended as there is high staff turnover and there are fewer midwives as expected in the maternity departments. Further to that, intensifying support and supervision and monitoring and evaluating the system on regular basis. There's need to conduct audits timely after a maternal death occurs when the memory is still fresh and learning on time to avoid future deaths of similar avoidable causes and to conduct audits for all maternal deaths in order not to miss opportunities to learn.

## Supporting information

**S1 Dataset.**
(XLS)

**S1 File.**
(DOCX)

## Acknowledgments

We would like to acknowledge Makonde District health team and Zimbabwe Field Epidemiology Training Programme for making our study a success.

## Author Contributions

**Conceptualization:** Tsitsi Brenda Makanyanga, Bernard Madzima, More Mungati, Addmore Chadambuka, Notion Tafara Gombe, Tsitsi Patience Juru, Chukwuma David Umeokonkwo, Mufuta Tshimanga.

**Formal analysis:** Tsitsi Brenda Makanyanga, Bernard Madzima, More Mungati, Addmore Chadambuka, Notion Tafara Gombe, Tsitsi Patience Juru, Chukwuma David Umeokonkwo.

**Investigation:** More Mungati, Addmore Chadambuka, Notion Tafara Gombe.

**Methodology:** Tsitsi Brenda Makanyanga, More Mungati, Chukwuma David Umeokonkwo, Mufuta Tshimanga.

**Resources:** Bernard Madzima, Mufuta Tshimanga.

**Supervision:** Bernard Madzima, Notion Tafara Gombe, Tsitsi Patience Juru, Chukwuma David Umeokonkwo, Mufuta Tshimanga.

**Writing – original draft:** Tsitsi Brenda Makanyanga, Bernard Madzima, More Mungati, Addmore Chadambuka, Notion Tafara Gombe, Tsitsi Patience Juru, Chukwuma David Umeokonkwo, Mufuta Tshimanga.

**Writing – review & editing:** Tsitsi Brenda Makanyanga, More Mungati, Addmore Chadambuka, Notion Tafara Gombe, Tsitsi Patience Juru, Chukwuma David Umeokonkwo, Mufuta Tshimanga.

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
