## [Decision Letter · Decision Letter 0]

2 May 2023

PONE-D-22-28650Maternal death surveillance and response system evaluation in Makonde District, Zimbabwe, 2021PLOS ONE

Dear Dr. Chadambuka,

Thank you for submitting your manuscript to PLOS ONE. After careful consideration, we feel that it has merit but does not fully meet PLOS ONE’s publication criteria as it currently stands.Thank you for considering PLoS ONE for your manuscript " Maternal death surveillance and response system evaluation in Makonde district, Zimbabwe, 2021". Peer review of your manuscript is now complete and, based on the reports, a major decision has been made. 

We look forward to receiving your revised manuscript.

Kind regards,

Richard Kalisa, MD, PhD

Academic Editor

PLOS ONE

2. Please include a copy of your questionnaire as a Supporting Information file or provide a link if it is available through an online repository

Additional Editor Comments:

**John Azaare, Ph.D. (Reviewer 1)**

Major decision

Abstract

Introduction

1. Your introductory statement “…with 810 women dying every day from preventable causes related to pregnancy” refers to figure as of 2017, five years behind. This (2017) should be refenced in the statement. Alternatively, the authors should revise the statement to report the current figures of maternal mortality globally and in sub-Saharan Africa and then in Zimbabwe.

2. The statement “Globally the maternal mortality ratio is declining however the average rate of decline is not fast enough to achieve SDG target of 70 maternal deaths per 100000 live births” is similar to the statement “The United Nations Sustainable Development Goal target 3.1 aims at reducing global maternal mortality ratio to less than 70 per 100 000 live births by 2030…” and conveys the same meaning. I suggest you consider merging the two.

3. The authors referenced “the Centre for Disease Control” which collaborated with the World Health Organization. Could you be specific about the particular Centre for Disease Control? We have so many Centres for Disease Control who could partner WHO to implement programmes in LMIC.

4. The authors explained that the “Maternal Death Surveillance and Response System (MDSR)” is what is partially being evaluated in Makonde District. Perhaps, the authors could be much clearer on the motivation for choosing the Makonde District. Although the authors explained that 62% of maternal death were notified and audited which presumably is low for them, this figure is left standing alone could rather be an impressive figure if compared contextually. Perhaps, it could be compared with the figures of other districts who implemented the “MDSR” to show a compelling reason for wanting to know the situation in the Makonde District in particular.

5. Also, note that “MDSR” used as an abbreviation in the manuscript does not exactly represent the “Maternal Death Surveillance and Response System (MDSR). Probably MDSRS. Please check and revise appropriately.

6. General Comment - The introductory aspects is generally scanty and does not address the topic area in literature, rational for the choice of study area and purpose of study, adequately.

Study setting

1. The authors gave statistics of antenatal care attendance in the district of study to include one visit etc. Again, we are not told what percentage made 8+ visits and if none, indicate so to put the reference data into proper perspective, especially as the WHO was part of the implementation of the MDSR.

2. The authors also left out health facility delivery percentages from the Makonde District, although they attempted to explain that health facilities existed widely with capabilities and with one referral centre – the Chinhoyi Provincial Hospital. Please add facility delivery figures of the district. It will be useful in your discussion.

Maternal Death Surveillance and Response System in Zimbabwe

1. The statement “…. health facilities and communities should include zero reporting in their routine reports” is not clear. Perhaps, you could explain what you mean by zero reporting.

2. Fig 1: Flow of information in a Maternal Death Surveillance and Response System in Zimbabwe is a good representation of information flow relative to the surveillance system. However, Mission Hospital/ Local Authorities is left standing alone with no link to source of information as though it is the final generator of information. It is not clear if the diagram is adopted or adapted or the original creation of the authors. Perhaps the authors could consider creating a link of information from the community to the mission hospital/local authority, since the community is the original and last source of maternal data.

3. Also, I don’t find an in-text reference to figure 1. which makes the figure appear not useful in the manuscript, but that is not the case. Please reference the figure in the write up appropriately.

Study design

1. “The updated CDC guidelines for 163 Surveillance System Evaluation” … was the updated version utilized in the Makonde District surveillance, otherwise, the authors may have to explain why they chose the updated version of CDC guidelines.

2. What is “T5” registers?

Sample size and data collection

1. “A minimum sample size of 61 was calculated”. This does not marry well with “we interviewed five doctors and forty nurses from the provincial hospital, two doctors and six nurses from the rural hospital, one nurse and one doctor from the private maternity clinic, ten nurses from the urban clinics and fourteen nurses from the rural clinics”. Clarity is necessary. Your sample calculation of the mortality record for desk top review must stand out clear from your choice of health care professionals for interviews.

2. Could the authors share the data collection tools?

Ethical consideration

1. Perhaps the authors could quote the approval code or number from the Mashonaland West Provincial Ethics Committee.

Results

1. Only 41% were midwives, although a large portion of the health professionals interviewed were nurses. The expectation is that midwives should be the main target group for interview in matters of maternal deaths and audit. The authors need to explain further why this was not the case. Perhaps numbers and availability and willingness, to add meaning to the results.

2. The writing of the results should be re-organised to give a good flow. For example, “Overall, 61 (77.2%) of the 248 healthcare workers displayed good knowledge of the MDSR system. Table 3 shows the results” should be the opening paragraph of that section and followed by other relevant portions of the results which the authors intend to communicate rather than make this statement in the ending paragraphs.

3. “Assuming a nurse’s salary of $400 USD and a distance of 120 km of the furthest health facility to the district hospital” what is the need for introducing a nurse’s salary in the cost component of the results write up? It appears tangential to the study discourse. In any case, the entire idea of cost does not add value to the study objective. It can be removed without a significant damage to the study results.

Discussion

1. Generally good-

2. The authors made a statement that “Healthcare workers in Makonde District presumed the MDSR system to be useful as they utilised the information generated from audits to improve maternal health service delivery and monitor maternal mortality” yet, the district doesn’t seem to achieve what is expected. Could there be some conflicting reasons??? This needs to come out clearly in the discussion.

Limitation of study

1. Although the study is useful, it is limited in many ways. For example, not targeting midwives more in the study, given they are the professional grouping mandated to handle pregnant women limits the validity of information obtained from the health workers. This has to be stated in the manuscript.

Conclusion and Recommendations

1. Your results show 87.3% know the importance of the tool, 91% know the timeliness importance, 97% thinks the system is useful, 88.6% are willing to continue with the surveillance system yet “lack of motivation” and “training gap” appeared in your conclusion? Readers will not understand this. The authors need to revise their conclusion based on the study results.

2. The authors rather gave general recommendation including training requirement of staff and also recommended the need for ambulance services to be effectively provided. I don’t find these as borne out of the study results/discussion, especially as the surveillance process did not involve ambulance system and the results suggest the staff had good knowledge of the surveillance system.

3. The author’s recommendations ought to be specific to the gaps identified in their evaluation or emanating from the discussion of the results. Specific and targeted.

**Saumya Nanda, MD in OBGYN, FICOG (Reviewer 2)**

The manuscript is technically sound descriptive cross-sectional study.

A 1-5 Likert scale was used to measure level of knowledge among health care workers. Data was captured using Epi Info 7.2.4 statistical package. This software was used to conduct 221 univariate analysis to generate frequencies, means and proportions which is appropriate for the study. Data underlying the findings are available.

The manuscript is written in standard English without ambiguity. No typographical or grammatical errors were detected. Few lines of the Introduction are similar to the introduction of the following publication and needs to be addressed.

Millimouno TM, Sidibé S, Delamou A, Bello KOA, Keugoung B, Dossou JP, Beavogui AH, Meessen B. Evaluation of the maternal deaths surveillance and response system at the health district level in Guinea in 2017 through digital communication tools. Reprod Health. 2019 Jan 18;16(1):5. doi: 10.1186/s12978-019-0671-3. PMID: 30658674; PMCID: PMC6339333.

The manuscript is loosely inspired by the following publication:

Maphosa, M., Juru, T. P., Masuka, N., Mungati, M., Gombe, N., Nsubuga, P., & Tshimanga, M. (2019). Evaluation of the Maternal Death Surveillance and response system in Hwange District, Zimbabwe, 2017. BMC Pregnancy and Childbirth, 19(1). https://doi.org/10.1186/s12884-019-2255-1.

There is profound similarity in the aim, study design and methodology of the submitted manuscript with the above publication.

You may also include additional comments for the author, including concerns about dual publication, research ethics, or publication ethics. (Please upload your review as an attachment if it exceeds 20,000 characters)

Reviewers' comments:

Reviewer's Responses to Questions

**Comments to the Author**

1. Is the manuscript technically sound, and do the data support the conclusions?

Reviewer #1: Partly

Reviewer #2: Yes

2. Has the statistical analysis been performed appropriately and rigorously? 

Reviewer #1: Yes

Reviewer #2: Yes

3. Have the authors made all data underlying the findings in their manuscript fully available?

Reviewer #1: No

Reviewer #2: Yes

4. Is the manuscript presented in an intelligible fashion and written in standard English?

Reviewer #1: Yes

Reviewer #2: Yes

5. Review Comments to the Author

Reviewer #1: The authors attempted to evaluate a very important surveillance programme on marternal mortality. Their attempt is largely descriptive and less of analysis of association or impact evalaution. It is also imperitive to note that the authors relied much on data from nurses and doctors and less from midwives, which raises questions of data validity, given midwives are the first point of call on matters of maternal mortality.

Reviewer #2: The manuscript is technically sound descriptive cross-sectional study. A 1-5 Likert scale was used to measure level of knowledge among health care workers. Data was captured using Epi Info 7.2.4 statistical package. This software was used to conduct 221 univariate analysis to generate frequencies, means and proportions which is appropriate for the study. Data underlying the findings are available. The manuscript is written in standard English without ambiguity. No typographical or grammatical errors were detected. Few lines of the Introduction are similar to the introduction of the following publication and needs to be addressed.

Millimouno TM, Sidibé S, Delamou A, Bello KOA, Keugoung B, Dossou JP, Beavogui AH, Meessen B. Evaluation of the maternal deaths surveillance and response system at the health district level in Guinea in 2017 through digital communication tools. Reprod Health. 2019 Jan 18;16(1):5. doi: 10.1186/s12978-019-0671-3. PMID: 30658674; PMCID: PMC6339333.

The manuscript is loosely inspired by the following publication:

Maphosa, M., Juru, T. P., Masuka, N., Mungati, M., Gombe, N., Nsubuga, P., & Tshimanga, M. (2019). Evaluation of the Maternal Death Surveillance and response system in Hwange District, Zimbabwe, 2017. BMC Pregnancy and Childbirth, 19(1). https://doi.org/10.1186/s12884-019-2255-1.

There is profound similarity in the aim, study design and methodology of the submitted manuscript with the above publication.

6. PLOS authors have the option to publish the peer review history of their article (what does this mean?). If published, this will include your full peer review and any attached files.

Reviewer #1: **Yes: **John Azaare, PhD

Reviewer #2: No

---

## [Author Response · Author response to Decision Letter 0]

22 Jun 2023

Dear Editor/Reviewers

RE: Responses to comments raised in our submitted manuscript ‘Maternal death surveillance and response system evaluation in Makonde District, Zimbabwe, 2021’

Thank you for reviewing our manuscript and for the constructive comments. Please find below our responses to the comments.

Introduction 

1. Your introductory statement “…with 810 women dying every day from preventable causes related to pregnancy” refers to figure as of 2017, five years behind. This (2017) should be referenced in the statement. Alternatively, the authors should revise the statement to report the current figures of maternal mortality globally and in sub-Saharan Africa and then in Zimbabwe. 

We have revised the statement and put current figures recorded by WHO for 2020. The current figures for maternal mortality are included in the introduction already. They have been rearranged.

2. The statement “Globally the maternal mortality ratio is declining however the average rate of decline is not fast enough to achieve SDG target of 70 maternal deaths per 100000 live births” is similar to the statement “The United Nations Sustainable Development Goal target 3.1 aims at reducing global maternal mortality ratio to less than 70 per 100 000 live births by 2030…” and conveys the same meaning. I suggest you consider merging the two. 

Thank you, we have merged the statements.

 3. The authors referenced “the Centre for Disease Control” which collaborated with the World Health Organization. Could you be specific about the particular Centre for Disease Control? We have so many Centres for Disease Control who could partner WHO to implement programmes in LMIC.

It is the US CDC Division of Reproductive Health. I have added it to the manuscript.

4. The authors explained that the “Maternal Death Surveillance and Response System (MDSR)” is what is partially being evaluated in Makonde District. Perhaps, the authors could be much clearer on the motivation for choosing the Makonde District. Although the authors explained that 62% of maternal death were notified and audited which presumably is low for them, this figure is left standing alone could rather be an impressive figure if compared contextually. Perhaps, it could be compared with the figures of other districts who implemented the “MDSR” to show a compelling reason for wanting to know the situation in the Makonde District in particular. 

The highest number of maternal deaths was recorded in Makonde District. The provincial referral hospital is in Makonde District where most of the complicated maternity cases are referred to. The remaining six districts in the province achieved 100% notification and auditing of maternal deaths save for one district which achieved 80% with the second highest number of maternal deaths. 

 Moreover, Makonde recorded the highest number of maternal deaths versus the highest deliveries. 

5. Also, note that “MDSR” used as an abbreviation in the manuscript does not exactly represent the “Maternal Death Surveillance and Response System (MDSR). Probably MDSRS. Please check and revise appropriately. 

Thank you for the comment. We have researched and it is abbreviated as MDSR. The word “system” is not part of the abbreviation. We have taken note of it in the manuscript. It is written as MDSR system.

6. General Comment - The introductory aspects is generally scanty and does not address the topic area in literature, rational for the choice of study area and purpose of study, adequately. 

We have added flesh to the introduction capturing some literature, rationale for choice of study area and purpose of study.

Study setting 

1. The authors gave statistics of antenatal care attendance in the district of study to include one visit etc. Again, we are not told what percentage made 8+ visits and if none, indicate so to put the reference data into proper perspective, especially as the WHO was part of the implementation of the MDSR.

Thank you we have included the statistics. None had attended eight or more ante-natal care visits stipulated by WHO.

 2. The authors also left out health facility delivery percentages from the Makonde District, although they attempted to explain that health facilities existed widely with capabilities and with one referral centre – the Chinhoyi Provincial Hospital. Please add facility delivery figures of the district. It will be useful in your discussion 

We have added the number of deliveries and maternal deaths of the different facilities in the study setting section.

Number of deliveries and maternal deaths in Makonde district health facilities offering maternity delivery services are as follows;

Chinhoyi provincial hospital 6920 deliveries and 22 maternal deaths 

Makonde Christian hospital 1915 and 0 deaths

Mtala council clinic 537 and 1 death 

Kanyaga Rural health Centre 408 and 0 deaths

St Ruperts mission hospital 314 and 0 deaths

Umboe rural health centre 314 and 0 deaths

Manyamba rural health centre 195 and 1 death

Sadoma council clinic 193 and 0 deaths 

Shackleton council clinic 193 and 0 deaths

Kenzamba rural health centre 173 and 0 deaths

Kamhonde rural health centre 171 and 0 deaths

Green Valley rural health centre 163 and 0 deaths

 Chitambo private hospital 155 and 0 deaths

River ranch rural health centre 139 and 0 deaths

Municipal urban clinics 26 and 0 deaths

Data has been presented as a table in the manuscript (Table 1)

Maternal Death Surveillance and Response System in Zimbabwe

1. The statement “…. health facilities and communities should include zero reporting in their routine reports” is not clear. Perhaps, you could explain what you mean by zero reporting

Zero reporting means reporting of the absence of an event under surveillance for the review period. A report with a zero for maternal deaths is thus submitted in this instance. We have explained in the manuscript.

2. Figure 1: Flow of information in a Maternal Death Surveillance and Response System in Zimbabwe is a good representation of information flow relative to the surveillance system. However, Mission Hospital/ Local Authorities is left standing alone with no link to source of information as though it is the final generator of information. It is not clear if the diagram is adopted or adapted or the original creation of the authors. Perhaps the authors could consider creating a link of information from the community to the mission hospital/local authority, since the community is the original and last source of maternal data. 

The flowchart now shows the connection between the community and mission and local council clinics.

3. Also, I don’t find an in-text reference to Figure 1. which makes the figure appear not useful in the manuscript, but that is not the case. Please reference the figure in the write up appropriately.

Thank you I have referenced Figure 1 in text. The flow of information in a Maternal Death Surveillance and Response system in Zimbabwe is shown in Figure 1. 

Study design 

1. “The updated CDC guidelines for 163 Surveillance System Evaluation” … was the updated version utilized in the Makonde District surveillance, otherwise, the authors may have to explain why they chose the updated version of CDC guidelines. 

The surveillance system in Zimbabwe as a whole is premised on satisfying attributes that are stipulated in the CDC guidelines. Any evaluation therefore seeks to check whether these attributes are being met. That is why we used the updated guidelines to evaluate the system in Makonde District.

2. What is “T5” registers? 

It is a register which summarises all health events captured in a month. However, we have removed the term T5 and described the type of register in the study design.

Sample size and data collection 

1. “A minimum sample size of 61 was calculated”. This does not marry well with “we interviewed five doctors and forty nurses from the provincial hospital, two doctors and six nurses from the rural hospital, one nurse and one doctor from the private maternity clinic, ten nurses from the urban clinics and fourteen nurses from the rural clinics”. Clarity is necessary. 

We calculated a minimum sample size of 61. Looking at the establishment of doctors and nurses in post, 8% were doctors and 92% were nurses which translates to interviewing at least 5 doctors and 56 nurses. We therefore managed to interview 8 doctors and 71 nurses over the period we carried out the study. Coming down to the facilities we also used proportions based on the establishment. From the provincial hospital we calculated at 31 nurses and 4 doctors, 14 nurses from the rural clinics, 6 nurses and 1 doctor from the rural hospitals and 5 nurses from the urban clinics. We managed to reach the samples sizes calculated for each facility however additional doctors and nurses were obtained from the referral hospital where most of the deaths and discrepancies were observed. 

2. Your sample calculation of the mortality record for desk top review must stand out clear from your choice of health care professionals for interviews.

All mortality records over the study period were reviewed. 

3. Could the authors share the data collection tools? 

Yes, we have shared the data collection tools

 Ethical consideration 1. 

Perhaps the authors could quote the approval code or number from the Mashonaland West Provincial Ethics Committee.

Thank you. The approval number is JFREC-NAC/3/22

Results 

1. Only 41% were midwives, although a large portion of the health professionals interviewed were nurses. The expectation is that midwives should be the main target group for interview in matters of maternal deaths and audit. The authors need to explain further why this was not the case. Perhaps numbers and availability and willingness, to add meaning to the results.

We have given reasons why we had only 41% midwives among the interviewed nurses in the discussion. The major reason was high staff turnover.

2. The writing of the results should be re-organised to give a good flow. For example, “Overall, 61 (77.2%) of the 248 healthcare workers displayed good knowledge of the MDSR system. Table 3 shows the results” should be the opening paragraph of that section and followed by other relevant portions of the results which the authors intend to communicate rather than make this statement in the ending paragraphs. 

We have rearranged the writing of the results. Thank you.

 3. “Assuming a nurse’s salary of $400 USD and a distance of 120 km of the furthest health facility to the district hospital” what is the need for introducing a nurse’s salary in the cost component of the results write up? It appears tangential to the study discourse. In any case, the entire idea of cost does not add value to the study objective. It can be removed without a significant damage to the study results. 

We agree with the reviewer and we have removed cost component in the results section

 Discussion 

1. Generally good 

Thank you

2. The authors made a statement that “Healthcare workers in Makonde District presumed the MDSR system to be useful as they utilised the information generated from audits to improve maternal health service delivery and monitor maternal mortality” yet, the district doesn’t seem to achieve what is expected. Could there be some conflicting reasons??? This needs to come out clearly in the discussion. 

We have indicated in the discussion that to some extent the system was useful as information generated from the system was utilised in the district to improve future outcomes however they were failing to achieve the expected and missing some opportunities that would be useful in improving the system.

Limitation of the study 

1. Although the study is useful, it is limited in many ways. For example, not targeting midwives more in the study, given they are the professional grouping mandated to handle pregnant women limits the validity of information obtained from the health workers. This has to be stated in the manuscript. 

Thank you for the guidance we have included the limitation.

Conclusion and recommendations 

1. Your results show 87.3% know the importance of the tool, 91% know the timeliness importance, 97% thinks the system is useful, 88.6% are willing to continue with the surveillance system yet “lack of motivation” and “training gap” appeared in your conclusion? Readers will not understand this. The authors need to revise their conclusion based on the study results. 

We have edited the conclusion not to include lack of motivation and training gaps. However recommending on going on job training and refreshers especially with the high staff turnover and having general nurses who are not midwives working in the maternity wards.

 2. The authors rather gave general recommendation including training requirement of staff and also recommended the need for ambulance services to be effectively provided. I don’t find these as borne out of the study results/discussion, especially as the surveillance process did not involve ambulance system and the results suggest the staff had good knowledge of the surveillance system. 

Thank you, we have revised the recommendations section and removed general training and included training of more midwives.

3. The author’s recommendations ought to be specific to the gaps identified in their evaluation or emanating from the discussion of the results. Specific and targeted. 

We have revised the recommendations section.

We have focused my recommendations on the following findings;

1. Fewer midwives and high staff turn over

2. Poor data management.

3. Timeliness of notification and audits

Reviewer 2 comments There is profound similarity in the aim, study design and methodology of the submitted manuscript with the above publication. You may also include additional comments for the author, including concerns about dual publication, research ethics, or publication ethics. (Please upload your review as an attachment if it exceeds 20,000 characters) 

We have reworked the study design and methodology in the manuscript to exclude similarities.

---

## [Editor Report · Decision Letter 1]

19 Jul 2023

PONE-D-22-28650R1Maternal death surveillance and response system evaluation in Makonde District, Zimbabwe, 2021PLOS ONE

Dear Dr. Chadambuka,

Thank you for submitting your manuscript to PLOS ONE. After careful consideration, we feel that it has merit but does not fully meet PLOS ONE’s publication criteria as it currently stands. Therefore, we invite you to submit a revised version of the manuscript that addresses the points raised during the review process.

We look forward to receiving your revised manuscript.

Kind regards,

Richard Kalisa, MD, PhD

Academic Editor

PLOS ONE

Journal Requirements:

**Additional Editor Comments:**

Thanks for addressing the reviewers comments. However, there are still minor revisions prior acceptance of your manuscript are listed below:

1.This manuscript should be proofread by a native English speaker

2. I noted the cost of notifying and investigating a single death was USD$65.18. Please clarify to your readers its relevance as it was ONLY mentioned in the abstract and not elsewhere.

3. The manuscript needs to formatted in accordance journal's manuscript style.

---

## [Author Response · Author response to Decision Letter 1]

5 Sep 2023

General 

This manuscript should be proofread by a native English speaker. 

Thank you. We have given it to two colleagues who have assisted with grammar, thank you.

Abstract 

I noted the cost of notifying and investigating a single death was USD$65.18. Please clarify to your readers its relevance as it was ONLY mentioned in the abstract and not elsewhere. 

Thank you for bringing this up. I have removed the cost aspect in the abstract. It was an omission, I deleted the cost aspect in the main manuscript as advised in the previous set of comments and forgot to delete the aspect in the abstract. 

General 

The manuscript needs to be formatted in accordance journal's manuscript style. 

Thank you. We have tried our best to format the manuscript to the journal's manuscript style.

---

## [Editor Report · Decision Letter 2]

6 Feb 2024

PONE-D-22-28650R2Maternal death surveillance and response system evaluation in Makonde District, Zimbabwe, 2021PLOS ONE

Dear Dr. Chadambuka,

Thank you for submitting your manuscript to PLOS ONE. After careful consideration, we feel that it has merit but does not fully meet PLOS ONE’s publication criteria as it currently stands. Therefore, we invite you to submit a revised version of the manuscript that addresses the points raised during the review process.

We look forward to receiving your revised manuscript.

Kind regards,

Johanna Pruller, PhD

Associate Editor

PLOS ONE

on behalf of 

Srinivasa Rao Gadde, Ph.D.

Academic Editor

PLOS ONE

Journal Requirements:

**Additional Editor Comments:**

Acceptance of the paper for publication is possible. Prior to accepting the submission, the following errors must be fixed by the authors:

Aline all the lines in the entire document.

Table contents and labels should be unbolded.

Examine the style of referring.

---

## [Author Response · Author response to Decision Letter 2]

19 Mar 2024

Responses to comments raised in our submitted manuscript ‘Maternal death surveillance and response system evaluation in Makonde District, Zimbabwe, 2021’

Thank you for reviewing our manuscript and for the constructive comments. Please find below our responses to the comments.

Comment 1:Align all the lines in the entire document.

We have aligned the document

Comment 2: Table contents and labels should be unbolded.

We have unbolded table contents and their labels

Comment 3: Examine the style of referring

We have revisited references and added missing information on some references and edited some

---

## [Editor Report · Decision Letter 3]

25 Mar 2024

Maternal death surveillance and response system evaluation in Makonde District, Zimbabwe, 2021

PONE-D-22-28650R3

Dear Dr. Chadambuka,

We’re pleased to inform you that your manuscript has been judged scientifically suitable for publication and will be formally accepted for publication once it meets all outstanding technical requirements.

Kind regards,

Srinivasa Rao Gadde, Ph.D.

Academic Editor

PLOS ONE

---

## [Editor Report · Acceptance letter]

29 Mar 2024

PONE-D-22-28650R3 

PLOS ONE

Dear Dr. Chadambuka, 

I'm pleased to inform you that your manuscript has been deemed suitable for publication in PLOS ONE. Congratulations! Your manuscript is now being handed over to our production team.

Kind regards, 

on behalf of

Professor Srinivasa Rao Gadde 

Academic Editor

PLOS ONE